# Continued versus discontinued oxytocin after the active phase of labor: An updated systematic review and meta-analysis

**Danni Jiang[1]☯, Yang Yang[2]☯, Xinxin Zhang[2], Xiaocui Nie[2]\***

**1** Graduate School, Dalian Medical University, Dalian, Liaoning, China, **2** Department of Gynecology, Shenyang Women's and Children's Hospital, Shenyang, Liaoning, China

☯ These authors contributed equally to this work.
\* xiaocui_nie@163.com

**Data Availability Statement:** All relevant data are within the paper and its Supporting Information files.

## Abstract

### Objective

To systematically assess the effect of discontinued vs continued oxytocin after active stage of labour is established.

### Methods

Pubmed, Embase, and the Cochrane Library were systematically searched to 18 April 2021. The risk ratio or mean difference with corresponding 95% confidence interval were computed to investigate the effect of intervention or control on maternal and fetus outcomes. This review was registered in the International Prospective Register of Systematic Reviews: CRD42021249635.

### Results

Discontinuing oxytocin when the active labour was established might decrease the risk of cesarean delivery [RR (95% CI): 0.84 (0.72–0.98), P = 0.02]. However, when we restricted our analysis to women who performed cesarean section after the active phase was reached, the difference was no longer significant [RR (95% CI): 0.82 (0.60–1.10), P = 0.19]. The incidence of uterine tachysystole [RR (95% CI): 0.36 (0.27–0.49)], postpartum hemorrhage [RR (95% CI): 0.78 (0.65–0.93)], and non-reassuring fetal heart rate [RR (95% CI): 0.66 (0.58–0.76)] were significantly lower in the oxytocin discontinuation group. We also found a possible decrease in the risk of chorioamnionitis in discontinued oxytocin group [RR (95% CI): 2.77 (1.02–5.08)]. An increased duration of active [MD (95% CI): 2.28 (2.86–41.71)] and second [MD (95% CI): 5.36 (3.18–7.54)] phase of labour was observed in discontinued oxytocin group, while the total delivery time was not significantly different [MD (95% CI): 20.17 (-24.92–65.26)].

### Conclusion

After the active labor is reached, discontinuation of oxytocin could be considered a new recommendation for the improved maternal and fetal outcomes without delaying labour.

**Funding:** The author(s) received no specific funding for this work.

**Competing interests:** The authors have declared that no competing interests exist.

## Introduction

Approximately one in every four term pregnant women have their labor induced [1–3]. Oxytocin, first synthesized in 1954, has been used widely for induction of labor [4]. Its application needs a balance between the effect on labor progression and the risk of uterine hyperstimulation, defined as more than five contractions in 10 minutes [5]. Frequent contractions, along with too short relaxation period, may lead to insufficient oxygen supplied to the fetus and abnormal fetal heart rate (FHR), which may require immediate intervention by instrumental assistance or cesarean delivery [6]. Also, oxytocin use may increase the maternal risk of uterine rupture and postpartum hemorrhage [7, 8]. A preliminary study indicates that oxytocin use raises the risk of unsuccessful breastfeeding [9]. Additionally, prolonging oxytocin use duration down-regulates the sensibility of oxytocin receptors in uterine myometrium, which lead to decreased efficiency of labour induction and increased the risks of maternal complications [10]. Few but existed, excessive oxytocin can cause seizure, hyponatremia, water retention, myocardial ischemia, and coma [11]. Furthermore, its use also has been doubted that whether oxytocin usage might have bad effect on children's behavior development over the long term since it can cross the placenta barrier [12].

Recently, oxytocin has gained increasing concern and is considered as 1 of the 12 most dangerous medications in hospital [13]. Although extensively used, no consensus exists regarding the optimal administration scheme, as well as the duration of oxytocin infusion. Some protocols for oxytocin administration have been studied, such as high versus low dose, pulsatile/intermittent versus continuous administration, automatic feedback system, and discontinuing of oxytocin when the active labour is established [14–18]. Of them, stopping oxytocin when the active stage of labour is established rather than continuing infusion until delivery gains growing interest and are often studied.

Four meta-analyses have been published on this issue and suggests that the labour will continue even if oxytocin stimulation is stopped [19–22]. However, related maternal and neonatal outcomes between continued oxytocin (OC) versus discontinued oxytocin (OD) are highly controversial and inconclusive. Whether discontinued oxytocin does have some advantages over the conventional administration therefore remains uncertain. In recent three years, some additional trails have been published concerning this important but inconclusive issue. This review and meta-analysis aims to systematically analyze and integrate the updated evidence on discontinued versus continued oxytocin for induction of labor and provide high-quality evidence for improving maternal and fetus outcomes as well as guiding future clinical practice.

## Methods

This review was done according to PRISMA (Preferred Reporting Items for Systematic Reviews and Meta-Analyses) [23]. This review was registered in the International Prospective Register of Systematic Reviews (PROSPERO): CRD42021249635.

### 1) Literature search

Literature searching was performed systematically of Pubmed, Embase, and the Cochrane Library to 18 April 2021 for relevant articles comparing discontinued versus continued oxytocin for the induction of labour without language restrictions. We combined MeSH terms 'oxytocin', 'Labor, Induced' with any related free terms for research searching. Manual search of reference lists from screened articles was also performed for other potential eligible studies.

## 2) Study selection

After removing the duplicate, the title and abstract of screened studies were assessed for potentially applicable articles. Then the full text of relevant studies were obtained and assessed, and the studies that accords with the eligibility criteria were included for this review. This process was independently conducted by two reviewers, any disagreement can be resolved by consulting with the third reviewer. The inclusion criteria were: (1) Study type: Randomized controlled trial (RCT); (2) Population: women who use oxytocin stimulation for induced labor; (3) Intervention: discontinuting oxytocin when the active stage of labor is established; (4) Comparison: continuting of oxytocin infusion till delivery. The exclusion criteria were: (1) abstracts, reviews, case reports, letters, or observational studies; (2) studies presented with insufficient original data, and (3) studies with duplicate data or repeated analysis.

## 3) Data extraction and quality assessment

Two reviewers were responsible for the data extraction process independently. If divergence occurred, the authors discussed it with the third reviewer to reach a consensus. The following items were extracted: (1) General information: the first author, publication year, country, sample size; (2) Study characteristics: study design, inclusion and exclusion criteria, definition of active phase, oxytocin regimen, protocol adherence; (3) Maternal and fetal outcomes: caesarean delivery, duration of the active stage of labor (mean ± SD), duration of the second stage of labor (mean ± SD), delivery time (mean ± SD), vaginal instrumental delivery, uterine tachysystole, epidural use, chorioamnionitis, third- or fourth-degree perineal tear, postpartum haemorrhage, exclusive breastfeeding at discharge, non-reassuring FHR, apgar score at 5 mint <7, arterial umbilical pH< 7.10, neonatal asphyxia, NICU admission.

Risk of bias assessment was performed independently by two reviewers in accordance with the risk of bias tool in Cochrane Handbook based on items below: random sequence generation; allocation concealment; blinding of participants, personnel, and outcome assessment; incomplete outcome data; selective reporting; other bias [24]. Each item was categorized as low, unclear or high risk of bias. Furthermore, two reviewers assessed the quality of the evidence for each outcome independently using GRADE (Grading of Recommendations, Assessment, Development, and Evaluation) approach [25], and the evidence was downgraded from 'high quality' for limitations based on the following items: study limitations, consistency of effect, imprecision, indirectness, and publication bias. For the quality assessment process, reviewers resolved the divergence by discussion or the arbitration of the third reviewer.

## 4) Data synthesis

Data synthesis process was performed using Review Manager 5.3 statistical software. The risk ratio (RR) or mean difference (MD) with their 95% confidence interval (CI) were calculated separately for dichotomous or continuous data. $I^2$ statistic was used for assessing heterogeneity, $I^2 \leq 50$ indicated an acceptable heterogeneity and a fixed-effect model was used to combine the data; or else, the random-effect model was used when a substantial heterogeneity ($I^2 > 50$) was detected across the studies. Publication bias across studies included in cesarean delivery analysis was assessed using Begg's and Egger's tests with Stata 14.0 software. The P-value of <0.05 was considered as significant statistically. Since no substantive heterogeneity was detected between primary outcome (cesarean delivery), none of subgroup analysis or sensitivity analysis were made.

## Results

### 1) Study selection

A total of 2606 studies were searched through electronic database and manual search. After removing duplicated documents, 2461 studies were screened by title and abstract. Then, 21 studies were further assessed by reading full-text, and 13 studies [18, 26–37] that in accordance with inclusion criteria were selected for this meta-analysis (Fig 1).

### 2) Study characteristics

The detailed characteristics of included studies are shown in Table 1, and the detailed oxytocin protocols in each study see Table 2. Protocol adherence for OD and OC Groups were shown in Table 3. A total of thirteen RCTs including 1696 in OD group and 1678 in OC group. These studies were published between 2004 and 2021. And the sample size of involved studies ranged from 90 to 1200 women. The definition of the active phase of labour concerning cervical dilation extent was 4 cm in 2 studies [30, 32], 5 cm in 7 studies [17, 18, 29, 33–35, 37], 6 cm in 1 study [28], 4–5 cm in 1 study [26], 4–6 in 1 study [31], and not defined in 1 study [36].

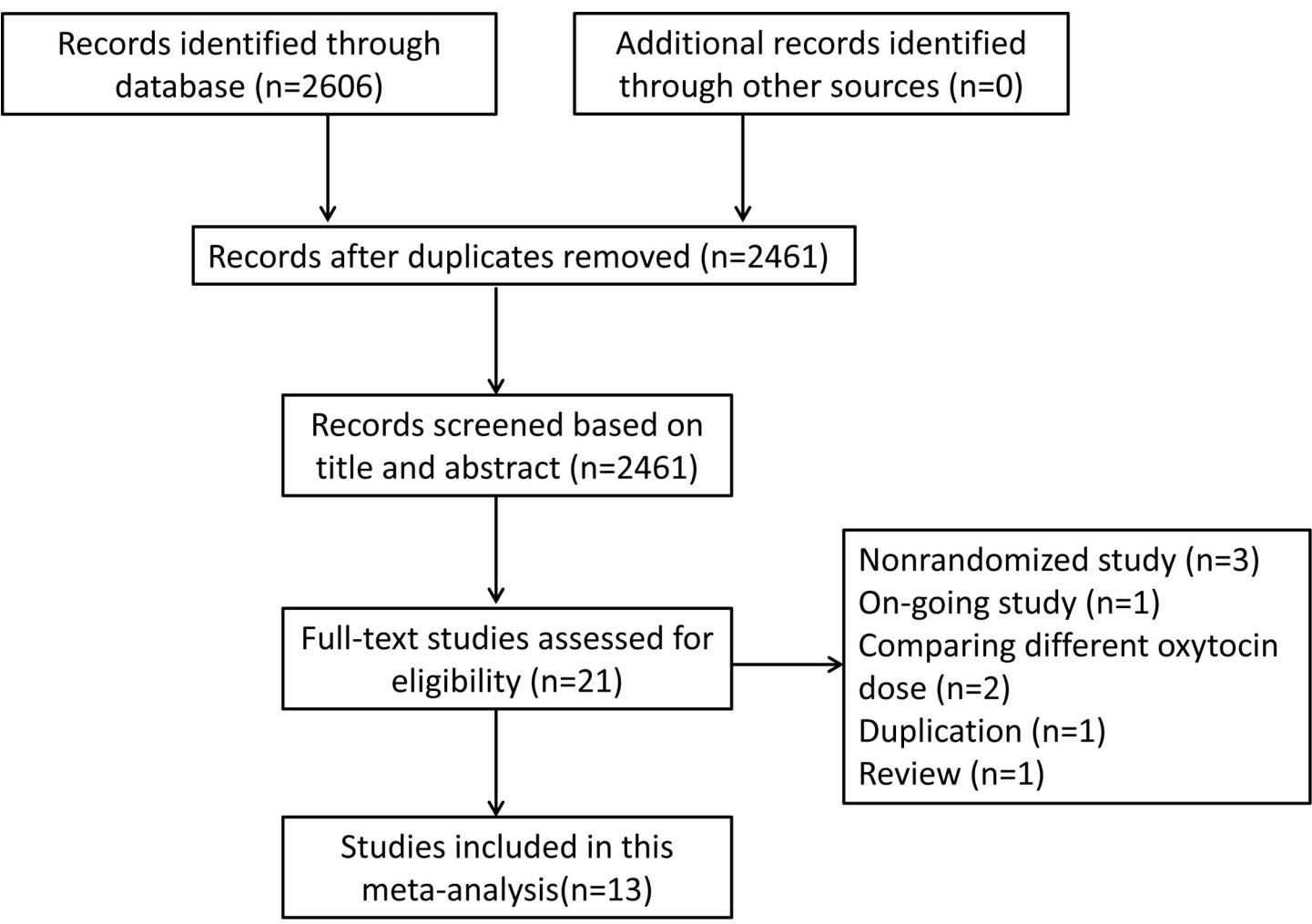

**Fig 1. The flow chart of literature selection process.**

**Table 1. Characteristics of the included studies.**

| Study year | Country | Sample size | Inclusion criteria | Defination of active phase | Intervention group | Control group |
|---|---|---|---|---|---|---|
| **Bahadoran 2011 [26]** | Iran | 104 | Singleton gestations ≥ 37 weeks, Bishop <5, BMI <26 | Dilatation of 4 cm and 80% effacement, or 5 cm without considering effacement | Oxytocin+500 cc of Ringer's solution | Oxytocin until delivery |
| **Begum 2013 [27]** | Bangladesh | 100 | Singleton gestations ≥ 37 weeks | Cervical dilatation of 5 cm | Oxytocin was discontinued after the active phase | Oxytocin until delivery on the same dose |
| **Bioe 2021 [28]** | Denmark and Netherlands | 1200 | Singleton gestations ≥ 37 weeks or PROM without progression in labour | Ruptured membranes with cervical dilatation of 6 cm and 100% effacement, and at least 3 contractions/10 minutes. | Oxytocin+500mL of 0.9% of NaCl solution | Oxytocin until delivery at the standard concentration |
| **Bor 2016 [29]** | Denmark | 200 | Singleton gestations ≥ 37 weeks, cervical dilation ≤ 4cm | Cervical dilatation of 5 cm | Oxytocin was discontinued after the active phase | Oxytocin until delivery on the same dose |
| **Chookijkul 2016 [30]** | Thailand | 340 | Singleton gestations ≥ 37 weeks, estimated fetal weight < 4000 g, Bishop > 4 | Cervical dilatation of 4 cm with good uterine contraction | Oxytocin+500mL of 0.9% of NaCl solution | Oxytocin until delivery |
| **Chopra 2015 [31]** | India | 106 | Singleton gestations ≥ 36 weeks | Cervical dilatation of 4–6 cm | Oxytocin+500mL of 0.9% of NaCl solution | Oxytocin until delivery on the same dose |
| **Daniel-Spiegel 2004 [18]** | Israel | 104 | Singleton gestations ≥41 weeks, or PROM >24 hours, or IUGR, or diabetes | Cervical dilatation of 5 cm | Oxytocin was discontinued after the active phase | Oxytocin until delivery on the same dose |
| **Diven 2012 [32]** | USA | 252 | Singleton gestations ≥ 36 weeks | Cervical dilatation of 4 cm, and regular contractions | Oxytocin was discontinued after the active phase | Oxytocin titrated to target 3–5 contractions /10 minutes until delivery |
| **Eissa 2019 [33]** | Egypt | 90 | Singleton gestation ≥ 37 weeks, postdated pregnancy (>42 weeks), Bishop ≥ 6, PROM at term, mild preeclampsia/ hypertension ≥ 39 weeks, oligohydramnios | Cervical dilatation of 5 cm | Oxytocin was discontinued after the active phase | Oxytocin until delivery on the same dose |
| **Mitra 2019 [34]** | India | 200 | Women ≥ 18 years old, primigravida, singleton gestations ≥ 37 weeks with cervical dilation ≤ 3cm and PROM | Cervical dilatation of 5 cm | Oxytocin+500mL of 0.9% of NaCl solution | Oxytocin until delivery on the same dose |
| **Ozturk 2015 [35]** | Turkey | 130 | Nulliparous singleton gestations ≥ 36 weeks | Cervical dilatation of 5 cm | Oxytocin was discontinued after the active phase | Oxytocin until delivery on the same dose |
| **Rashwan 2011 [36]** | Egypt | 200 | Singleton gestations ≥ 37 weeks, Bishop >4 | Not defined | Oxytocin was discontinued after the active phase | Oxytocin until delivery on the same dose |
| **Ustunyurt 2007 [37]** | Turkey | 342 | Singleton gestations ≥ 37 weeks | Cervical dilatation of 5 cm, and regular contractions at 3 min intervals | Oxytocin+500mL of 0.9% of NaCl solution | Oxytocin until delivery on the same dose |

Women of the OD group received placebo with saline/ringer's solution in 5 studies [26, 28, 30, 31, 37], while other studies simply discontinued the oxytocin inclusion once active labor was established. In the OC group, oxytocin was used until delivery on the same dose in 9 studies [18, 27, 29, 31, 33–37], used at the standard concentration in 1 study [28], used titrated to target 3–5 contractions/10 minutes in 1 study [32], and 2 studies did not specify oxytocin dose [26, 30]. The publication bias for primary outcome (cesarean section) was observed among the twelve studies with the p-value of 0, see the funnel plot (Fig 2). The risk of bias assessment result for this systematic review is shown in Fig 3 Risk of bias summary and Fig 4 Risk of bias graph. And the details for GRADE assessment are presented in Table 4. The GRADE assessment for each outcome ranged from very low certainty to moderate certainty.

**Table 2. Detailed oxytocin protocol in each study.**

| | Oxytocin dilution | Starting dose | Increasing dose | Maximal dose |
|---|---|---|---|---|
| **Bahadoran 2011** | 5 IU in 500 mL of Ringer's solution | 6 mIU/minute | 6 mIU/30 min until regular contractions[a] | NR[b] |
| **Begum 2013** | 5 IU in 500 mL of Ringer's solution or 2.5 IU in 5% dextrose | 10 droups/minute | Until regular contractions[a] | 20 mIU/min |
| **Bioe 2021** | 10 IU in 1000 mL of 0.9% NaCl or 5 IU in 50 mL of 0.9% NaCl | 3.3 mIU/min | 3.3 mIU/20 min until regular contractions[a] | 30/33 mIU/ min in Denmark / Netherlands |
| **Bor 2016** | 5 IU in 500 mL of 0.9% NaCl | 3.3 mIU/minute | 3.3 mIU/20 min until regular contractions[a] | 30 mIU/min |
| **Chookijkul 2016** | NR[b] | NR[b] | NR[b] | NR[b] |
| **Chopra 2015** | NR[b] | 3 mIU/minute | 3.3 mIU/30 min until regular contractions[a] | 42 mIU/min |
| **Daniel-Spiegel 2004** | 5 IU in 500 mL of 0.9% NaCl | 1 mIU/minute | 1 mIU/20 min until regular contractions[a] | 20 mIU/min |
| **Diven 2012** | 30 IU in 500 mL of 0.9% NaCl | NR[b] | Until regular contractions[a] | NR[b] |
| **Eissa 2019** | 5 IU in 500 mL of Ringer's solution | 2 mIU/minute | 2 drops/20 min until 3–4 contractions/10min | 20 mIU/min |
| **Mitra 2019** | 1.5 IU in 500 mL of 0.9% NaCl | 3 mIU/minute | 3 mIU/30 min until 3–4 contractions/10min | 15 mIU/min |
| **Ozturk 2015** | Oxytocin in 0.9% NaCl at a 1% concentration | 1–2 mIU/minute | 2 mIU/15 min until regular contractions[a] | 40 mIU/min |
| **Rashwan 2011** | 5 IU in 500 mL of 0.9% NaCl | 1 mIU/minute | 1 mIU/20 min until regular contractions[a] | 20 mIU/min |
| **Ustunyurt 2007** | 5 IU in 500 mL of 0.9% NaCl | 2 mIU/minute | 2 mIU/15 min until regular contractions[a] | NR[b] |

[a]Regular contractions: 3–5 contractions per 10 min;
[b]NR: not reported.

## 3) Meta-analysis

**3.1 Cesarean section.** Totally, 12 studies with 3270 women that explored the association between discontinued oxytocin and cesarean section risk were analyzed (Fig 5). Discontinuing oxytocin infusion after the active labor may decrease the risk of cesarean delivery [RR (95% CI): 0.84 (0.72–0.98), P = 0.02]. And no statistical heterogeneity was detected. Notably, when we restricted our analysis to women who performed cesarean section after the active phase was reached, the difference was no longer significant [RR (95% CI): 0.82 (0.60–1.10), P = 0.19] (Fig 6). Furthermore, no intergroup differences were detected concerning the indication for cesarean delivery such as fetal distress, non-reassuring FHR, arrest of labour, dystocia, chorioamnionitis, and suspicion of uterine rupture (Table 5).

**3.2 Maternal outcomes.** Six studies including 2015 women reported the correlation between discontinued oxytocin and uterine tachysystole (Fig 7). A significant decrease in the uterine tachysystole risk was observed in the OD group versus the OC group [RR (95% CI): 0.36 (0.27–0.49), P<0.00001]. We found no evidence of statistical heterogeneity. To assess postpartum hemorrhage, eight studies including 2484 patients were employed (Fig 8). Its risk reduced significantly in the discontinued oxytocin group [RR (95% CI): 0.78 (0.65–0.93), P = 0.006]. No significant heterogeneity was found across included studies. Two studies including 1450 patients explored the effect of discontinued oxytocin on chorioamnionitis, and we found a possible increased risk of chorioamnionitis in OD group without significant heterogeneity [RR (95% CI): 2.27 (1.02–5.08), P = 0.05] (Fig 9). However, no intergroup differences were found for other maternal outcomes such as vaginal instrumental delivery, epidural

**Table 3. Protocol adherence for OD and OC groups.**

| Study | Failure Protocol in OD[a] group | N (%) with failure protocol | Failure Protocol in OC[b] group | N (%) with failure protocol |
|---|---|---|---|---|
| Bahadoran [26] | NR[c] | NR[c] | NR[c] | NR[c] |
| Begum [27] | NR[c] | 0/50 | NR[c] | 0/50 |
| Bioe [28] | Slow labour progress, uterine tachysystole, FHR abnormalities, and others | 311/607 (51.2%) had oxytocin restarted | Slow labour progress, uterine tachysystole, FHR abnormalities, and others | 197/591 (33.3%) had oxytocin discontinued |
| Bor [29] | No cervical dilatation in two hours | 36/100 (36.0%) had oxytocin restarted | Non-reassuring FHR, or others | 3/100 (3.0%) had oxytocin discontinued |
| Chookijkul [30] | Poor uterine contraction | 16/170 (9.4%) had oxytocin restarted | Non-reassuring FHR | 15/170 (8.8%) had oxytocin discontinued |
| Chopra [31] | Inadequate uterine contractions (<3/10min) for two hours or more, or if cervical dilation did not improve | NR[c] | Non-reassuring FHR, or others | NR[c] |
| Daniel-Spiegel [18] | Inadequate uterine contractions for two hours or more, or if cervical dilation did not improve | 4/52 (7.7%) had oxytocin restarted | Non-reassuring FHR | 4/52 (7.7%) had oxytocin discontinued |
| Diven [32] | Lack of cervical change, or decrease in contraction frequency | 89/125 (71.2%): - 31/125 not discontinued despite randomization—58/125 had oxytocin restarted | Non-reassuring FHR, or others | 0/127 |
| Eissa [33] | NR[c] | NR[c] | NR[c] | NR[c] |
| Mitra [34] | Arrest of labour | 7/100 (7%) had oxytocin restarted | Uterine tachysystole, fetal distress | 15/100 (15%) had oxytocin discontinued |
| Ozturk [35] | Lack of cervical change, or decrease in contraction frequency | 0/66 | Non-reassuring FHR, or others | 0/64 |
| Rashwan [36] | NR[c] | NR[c] | NR[c] | NR[c] |
| Ustunyurt [37] | No cervical change for two hours despite adequate contractions | 11/168 (6.5%) had oxytocin restarted | Non-reassuring FHR | 8/174 (4.6%) had oxytocin discontinued |

[a]OD: discontinued oxytocin;

[b]OC: continued oxytocin;

[c]NR: not reported

use, third- or fourth-degree perineal tear, and exclusive breastfeeding at discharge, see pooled Table 6.

**3.3 Neonatal outcomes.** Eleven studies with a total of 2893 women reported the non-reassuring FHR result in discontinued versus continued oxytocin group (Fig 10). The pooled result demonstrated that the risk of non-reassuring FHR significantly decreased in the discontinued group [RR (95% CI): 0.66 (0.58–0.76), P<0.00001]. And we found no evidence of statistical heterogeneity. Other neonatal outcomes including Apgar score at 5 mint <7, arterial umbilical pH< 7.10, neonatal asphyxia, and NICU admission are not statistically different between groups (Table 6).

**3.4 Induction delivery interval.** The association between discontinued oxytocin infusion and the duration of active labor was analyzed in 9 studies involving 1536 patients (Fig 11). The pooled result indicated that discontinued may prolong the active stage of labor compared to continued oxytocin [MD (95% CI): 22.28 (2.86–41.71), P = 0.02]. The heterogeneity was statistically significant, then a random-effect modal was used. Similarly, the duration of the second labor was prolonged when the oxytocin is discontinued compared to continued group [MD (95% CI): 5.36 (3.18–7.54), P<0.00001, 8 trails, 2412 women] (Fig 12). The fixed-effect modal was chosen since no substantial heterogeneity was found. However, total delivery duration was

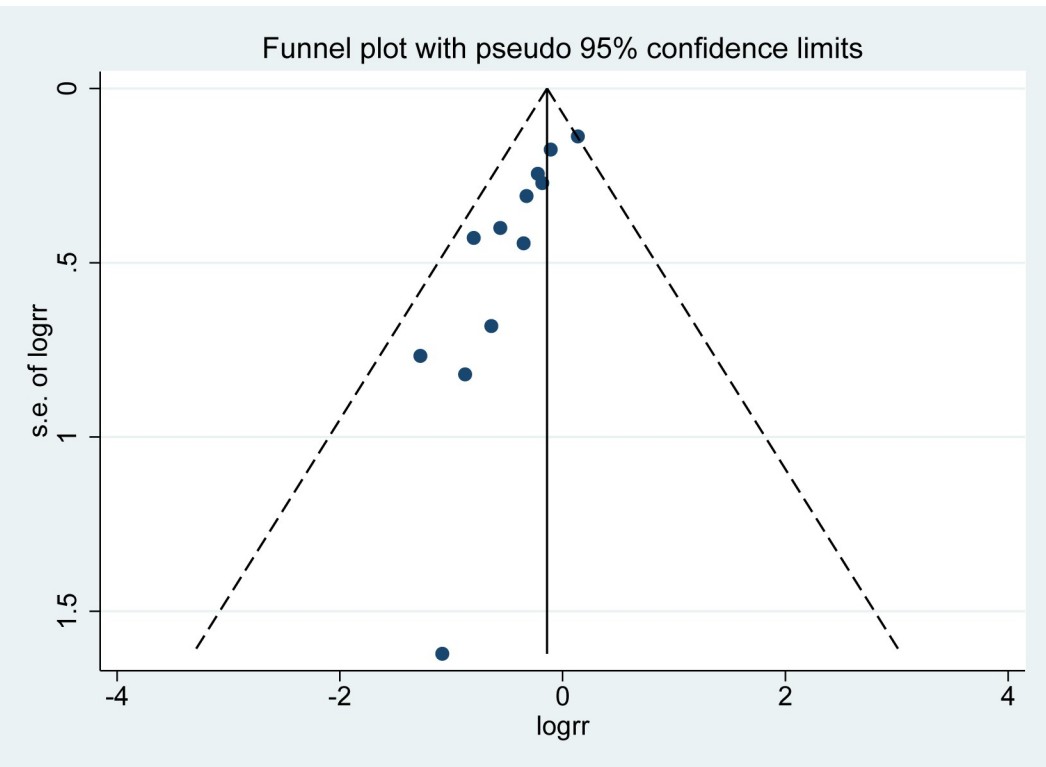

**Fig 2. The funnel plot of studies included in cesarean section analysis.**

not significantly different between discontinued and continued oxytocin groups [MD (95% CI): 20.17 (-24.92–65.26), P = 0.38, 6 trails, 2090 women] (Fig 13). The detected heterogeneity was significant, so we used a random-effect model for pooling the result.

## Discussion

### 1) Main findings and interpretation

13 RCTs with 3374 women were included in this review for comparing OD and OC in the induction of labour after the active phase was reached. The pooled results indicated that the discontinued oxytocin could reduce the risk of cesarean section, uterine tachysystole, postpartum hemorrhage, and non-reassuring FHR, without evidence of increasing the maternal and neonatal complications. Surprisingly, we observed an uncertain effect of discontinuation of oxytocin on the risk of maternal chorioamnionitis. Furthermore, the duration of the active and second stage of labor was significantly increased in OD group, yet the total delivery time was similar between groups.

### 2) Comparison with other studies

On this issue, 3 meta-analysis and 1 Cochrane systematic review has been published until now [19–22]. Vlachos, Saccone, and Hernández-Martínez et al. [19–21] found that discontinued oxytocin could significantly decrease the risk of cesarean delivery and uterine tachysystole. Our pooled result indicated that when the analysis was restricted in the women who really reached the active phase, the effect was not evident anymore. And this result coincides with a previous review by Boie et al. [22], yet our meta-analysis includes three most recent

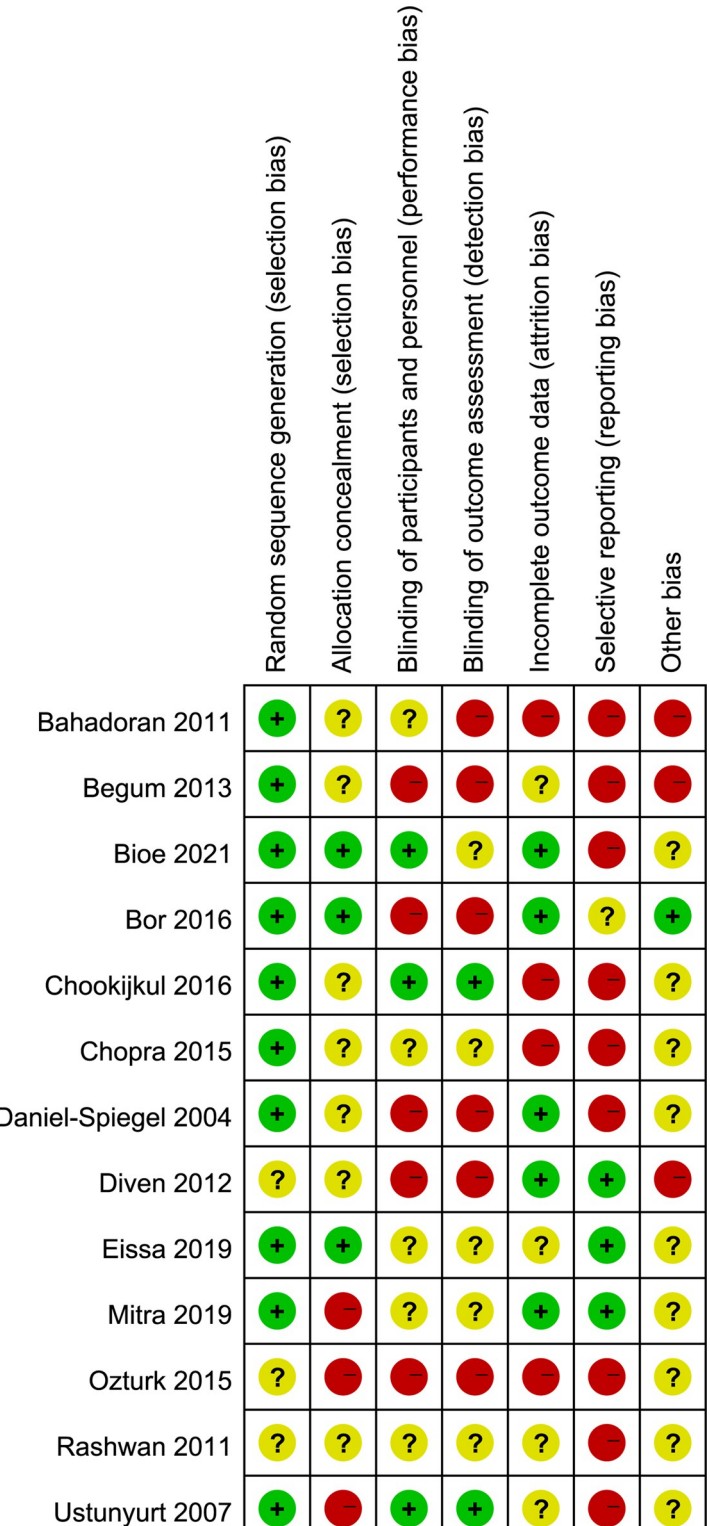

**Fig 3. Risk of bias summary.**

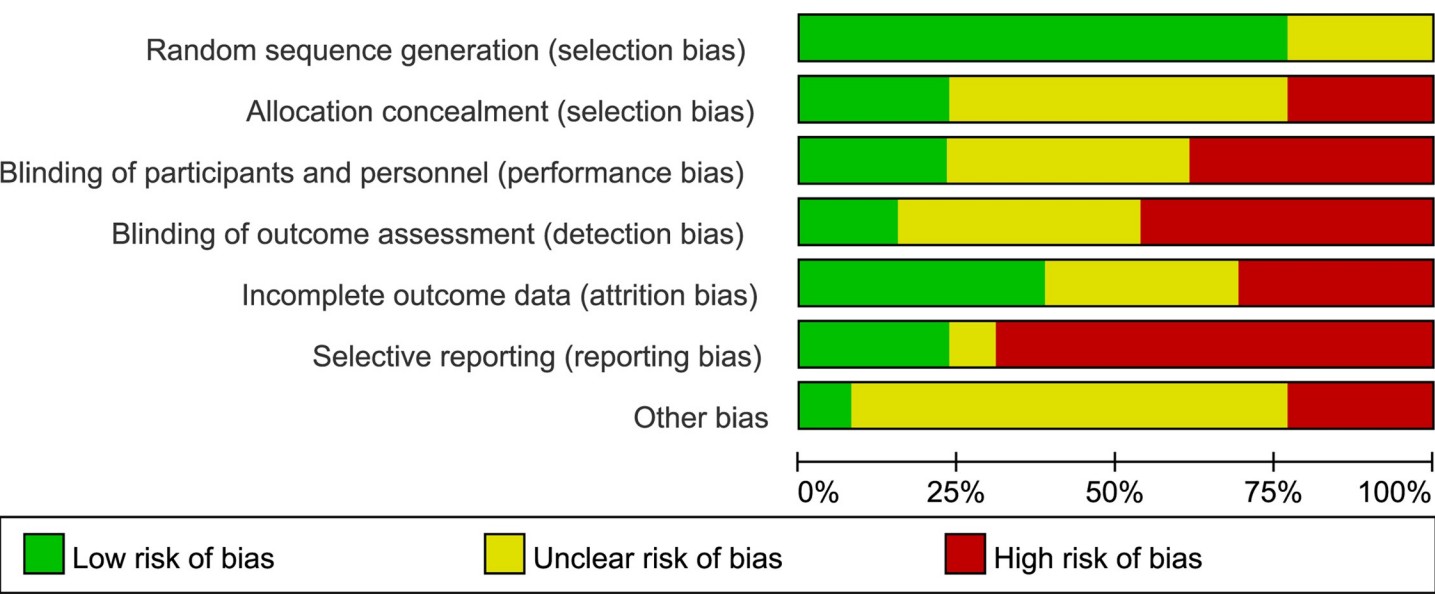

**Fig 4. Risk of bias graph.**

RCTs with 79% more women. Moreover, we found a decreased non-reassuring FHR risk in the OD group, which is opposite to the conclusion by Saccone et al. [20]. In our review, discontinued oxytocin increased the duration of active labor, while Vlachos and Hernández-Martínez et al. [19, 21] found no differences. The duration of second labor was also observed increased in OD group, which is not coincided with Saccone et al. [20]. In addition, the total delivery time was not different between groups, consist with previous reviews

**Table 4. Details for GRADE assessment.**

| Outcome | Quality | | | | | | Quality |
|---|---|---|---|---|---|---|---|
| | Design | Limitations | Inconsistency | Indirectness | Imprecision | Others | |
| Caesarean delivery | randomised trials | very serious | no serious | no serious | no serious | reporting bias | Very low |
| Duration of the active phase of labour | randomised trials | very serious | very serious | no serious | no serious | none | Very low |
| Duration of the second phase of labour | randomised trials | very serious | no serious | no serious | no serious | none | Low |
| Delivery time | randomised trials | serious | very serious | no serious | no serious | none | Very low |
| Vaginal instrumental delivery | randomised trials | serious | no serious | no serious | serious | none | Low |
| Uterine tachysystole | randomised trials | serious | no serious | no serious | no serious | none | Moderate |
| Epidural use | randomised trials | serious | serious | no serious | serious | none | Very low |
| Chorioamnionitis | randomised trials | serious | no serious | serious | no serious | none | Low |
| Third- or fourth-degree perineal tear | randomised trials | no serious | no serious | no serious | serious | none | Moderate |
| Postpartum haemorrhage | randomised trials | serious | no serious | no serious | no serious | none | Moderate |
| Exclusive breast feeding at discharge | randomised trials | no serious | very serious | no serious | serious | none | Very low |
| Non-reassuring FHR | randomised trials | very serious | no serious | no serious | no serious | none | Low |
| Apgar score at 5 mint <7 | randomised trials | serious | no serious | no serious | serious | none | Low |
| Arterial umbilical pH< 7.10 | randomised trials | serious | no serious | no serious | serious | none | Low |
| Neonatal asphyxia | randomised trials | no serious | no serious | no serious | serious | none | Moderate |
| NICU admission | randomised trials | very serious | no serious | no serious | serious | none | Very low |

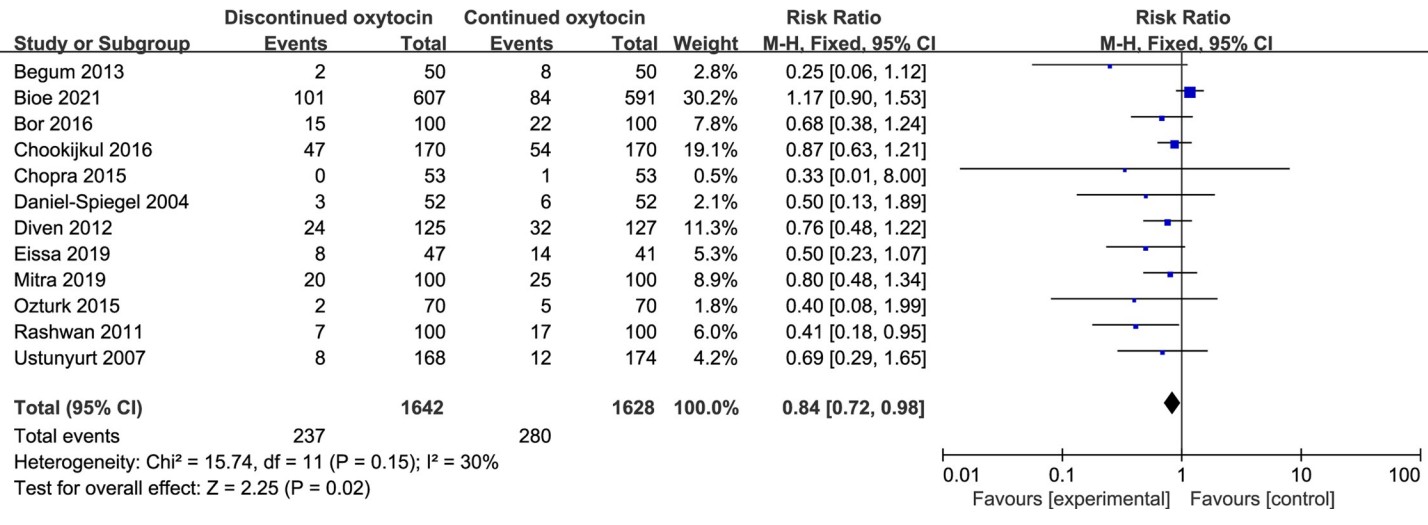

**Fig 5. Pooled results for each outcome.**

by Vlachos and Hernández-Martínez et al. [19, 21]. In contrast to Boie et al. [22], our review found an increased risk for chorioamnionitis in the discontinued oxytocin group. Of note, the p-value of 0.5 indicated that the conclusion should be interpreted with caution. And, we reported two additional outcomes compared to previous meta-analysis [19–22], the risk of neonatal asphyxia and exclusive breastfeeding at discharge were not significantly different between groups.

### 3) Strengths and limitations

One main strength of this review is that three recent trails are included with approximately 79% more women [22]. Additionally, the publication bias for the main outcome was also analyzed. The main limitation is the high heterogeneity of inclusion criteria and oxytocin protocol across included studies. Since no standard oxytocin scheme exists, starting dose, increasing dose, dilation from which oxytocin was discontinued, criteria for reestablish oxytocin and cesarean section were somewhat different between trails. And the subgroup analysis stratified by parity was not possible due to the limited data. Also, we observed a publication bias for the main result (cesarean section). In the future, includes more studies might be valuable, for example an ongoing trail in 20 maternity units in France with 2475 women [38].

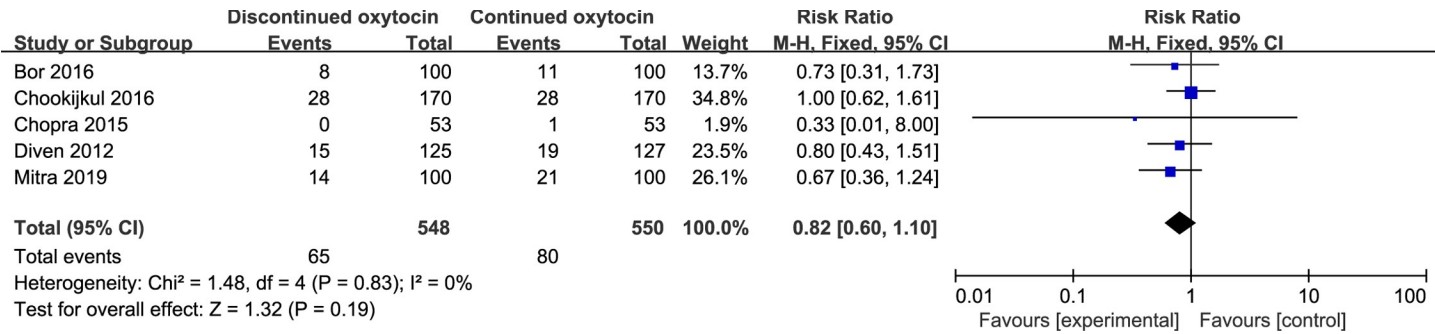

**Fig 6. Pooled results for each outcome.**

**Table 5. The analysis of indication for cesarean delivery.**

| Aurthor | Fetal distress | | Non-reassuring FHR[a] | | Arrest of labour | | Dystocia | | Chorioamnionitis | | Suspition of uterine rupture | |
|---|---|---|---|---|---|---|---|---|---|---|---|---|
| | OD[b] | OC[c] | OD[b] | OC[c] | OD[b] | OC[c] | OD[b] | OC[c] | OD[b] | OC[c] | OD[b] | OC[c] |
| Bioe 2021 | 26/607 | 13/591 | NR[d] | NR[d] | NR[d] | NR[d] | 62/607 | 58/591 | 2/607 | 1/591 | 3/607 | 1/591 |
| Bor 2016 | 3/100 | 7/100 | 2/100 | 3/100 | 12/100 | 15/100 | 12/100 | 15/100 | 1/100 | 0/100 | 0/100 | 2/100 |
| Chookijkul 2016 | NR[d] | NR[d] | 12/47 | 13/54 | NR[d] | NR[d] | NR[d] | NR[d] | NR[d] | NR[d] | NR[d] | NR[d] |
| Chopra 2015 | 0/53 | 0/53 | NR[d] | NR[d] | 0/51 | 1/53 | NR[d] | NR[d] | NR[d] | NR[d] | NR[d] | NR[d] |
| Daniel-Spiegel 2004 | NR[d] | NR[d] | 1/52 | 3/52 | 2/52 | 3/52 | NR[d] | NR[d] | NR[d] | NR[d] | NR[d] | NR[d] |
| Diven 2012 | 7/125 | 8/127 | 7/125 | 8/127 | 6/125 | 7/127 | NR[d] | NR[d] | NR[d] | NR[d] | NR[d] | NR[d] |
| Eissa 2019 | NR | NR | 3/47 | 7/41 | 2/47 | 3/41 | NR[d] | NR[d] | NR[d] | NR[d] | NR[d] | NR[d] |
| Mitra 2019 | NR | NR | 10/100 | 10/100 | 4/100 | 5/100 | NR[d] | NR[d] | 3/100 | 4/100 | NR[d] | NR[d] |
| Ozturk 2015 | 2/66 | 2/64 | NR[d] | NR[d] | 0/66 | 1/64 | NR[d] | NR[d] | NR[d] | NR[d] | NR[d] | NR[d] |
| Ustunyurt 2007 | 4/168 | 6/174 | NR[d] | NR[d] | 4/168 | 6/174 | NR[d] | NR[d] | NR[d] | NR[d] | NR[d] | NR[d] |
| No. of events | 42/1119 | 36/1109 | 35/471 | 44/474 | 30/709 | 41/711 | 74/707 | 73/691 | 6/807 | 5/791 | 3/707 | 3/691 |
| No. of studies | 6 | | 6 | | 8 | | 2 | | 3 | | 2 | |
| Overall effect (P-value) | 0.51 | | 0.35 | | 0.18 | | 0.96 | | 0.78 | | 0.98 | |
| RR 95%CI | 1.16 [0.75, 1.80] | | 0.82 [0.54, 1.24] | | 0.74 [0.47, 1.15] | | 0.99 [0.73, 1.35] | | 1.17 [0.38, 3.61] | | 0.98 [0.22, 4.34] | |

[a]FHR: fetal heart rate;

[b]OD: discontinued oxytocin;

[c]OC: continued oxytocin;

[d]NR: not reported

## 4) Implication

Exploring the optimal scheme for the management of induced labour is of vital importance for improving health outcomes for both the women and infants. Since synthesized in 1954, oxytocin has been the most widely used agent for inducing of labour [4]. Though extensively used, its ideal regimen regarding initial dose, increments, and maximal dose is still a lack of consensus [17, 39, 40]. An important issue to consider is that the use of oxytocin for induced labor is accompanied by an adverse effect of uterine hyperstimulation, which may causes adverse outcomes for maternal and fetus [6–12]. A long episode of induced labor is associated with uterine atony and postpartum bleeding that can not be stopped by oxytocin, which may caused by desensitized oxytocin receptor [41]. There is evidence indicates that the oxytocin of more than 10 hours' use may have no effect or even adverse effect on uterine

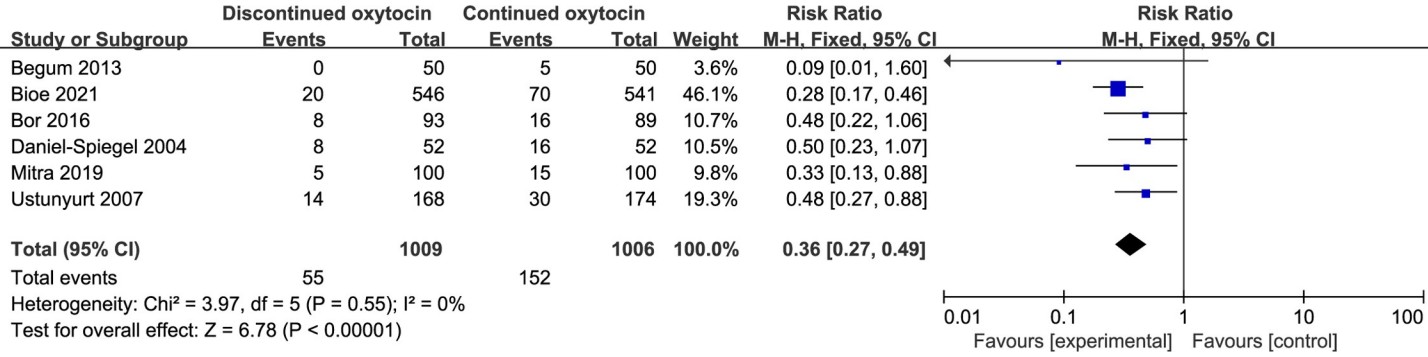

**Fig 7. Pooled results for each outcome.**

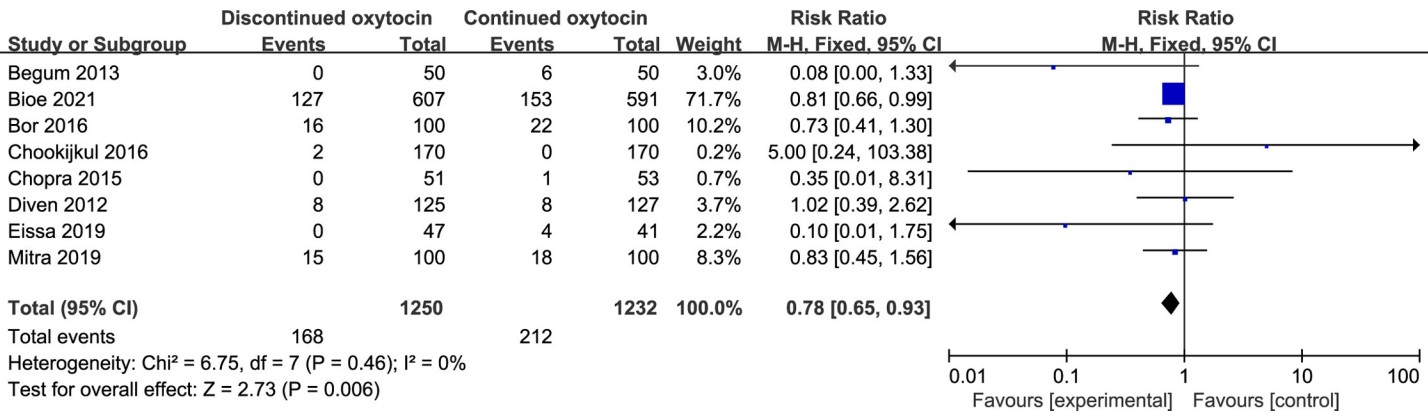

**Fig 8. Pooled results for each outcome.**

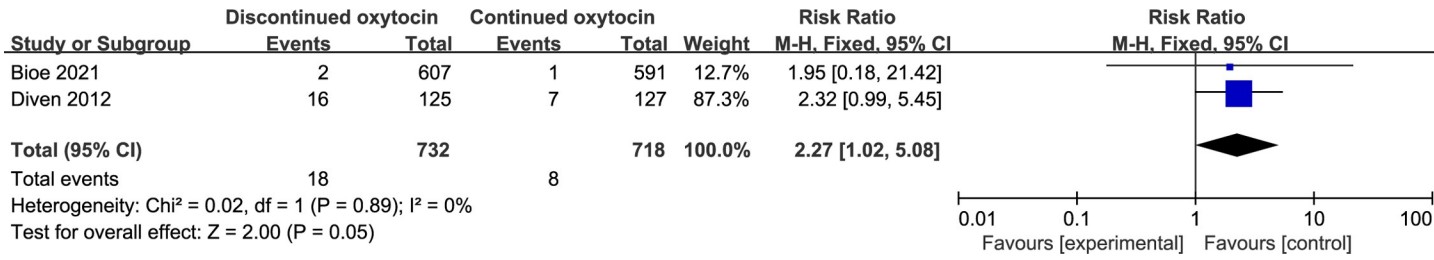

**Fig 9. Pooled results for each outcome.**

**Table 6. Pooled outcomes for this meta-analysis.**

| Outcome | No. of studies | No. of patients | Statistical method | Effect size | GRADE summary |
|---|---|---|---|---|---|
| 1 Caesarean delivery | 12 | 3270 | Risk Ratio (M-H, Fixed, 95% CI) | 0.84 [0.72,0.98] | Very low |
| 2 Duration of the active phase of labour | 9 | 1536 | Mean Difference (IV, Random, 95% CI) | 22.28 [2.86, 41.71] | Very low |
| 3 Duration of the second phase of labour | 8 | 2412 | Mean Difference (IV, Fixed, 95% CI) | 5.36 [3.18, 7.54] | Low |
| 4 Delivery time | 6 | 2090 | Mean Difference (IV, Random, 95% CI) | 20.17 [-24.92, 65.26] | Very low |
| 5 Vaginal instrumental delivery | 7 | 2246 | Risk Ratio (M-H, Fixed, 95% CI) | 0.93 [0.72, 1.20] | Low |
| 6 Uterine tachysystole | 6 | 2015 | Risk Ratio (M-H, Fixed, 95% CI) | 0.36 [0.27, 0.49] | Moderate |
| 7 Epidural use | 4 | 1754 | Risk Ratio (M-H, Random, 95% CI) | 1.04 [0.93, 1.17] | Very low |
| 8 Chorioamnionitis | 2 | 1450 | Risk Ratio (M-H, Fixed, 95% CI) | 2.27 [1.02, 5.08] | Low |
| 9 Third- or fourth-degree perineal tear | 2 | 1212 | Risk Ratio (M-H, Fixed, 95% CI) | 0.84 [0.51, 1.40] | Moderate |
| 10 Postpartum haemorrhage | 8 | 2482 | Risk Ratio (M-H, Fixed, 95% CI) | 0.78 [0.65, 0.93] | Moderate |
| 11 Exclusive breast feeding at discharge | 1 | 1158 | Risk Ratio (M-H, Random, 95% CI) | 1.03 [0.95, 1.12] | Very low |
| 12 Non-reassuring FHR | 11 | 2893 | Risk Ratio (M-H, Fixed, 95% CI) | 0.66 [0.58, 0.76] | Low |
| 13 Apgar score at 5 mint <7 | 6 | 2291 | Risk Ratio (M-H, Fixed, 95% CI) | 0.94 [0.46, 1.94] | Low |
| 14 Arterial umbilical pH< 7.10 | 5 | 2071 | Risk Ratio (M-H, Fixed, 95% CI) | 1.02 [0.71, 1.47] | Low |
| 15 Neonatal asphyxia | 2 | 1298 | Risk Ratio (M-H, Fixed, 95% CI) | 0.57 [0.17, 1.87] | Moderate |
| 16 NICU admission | 9 | 2920 | Risk Ratio (M-H, Fixed, 95% CI) | 0.81 [0.63, 1.06] | Very low |

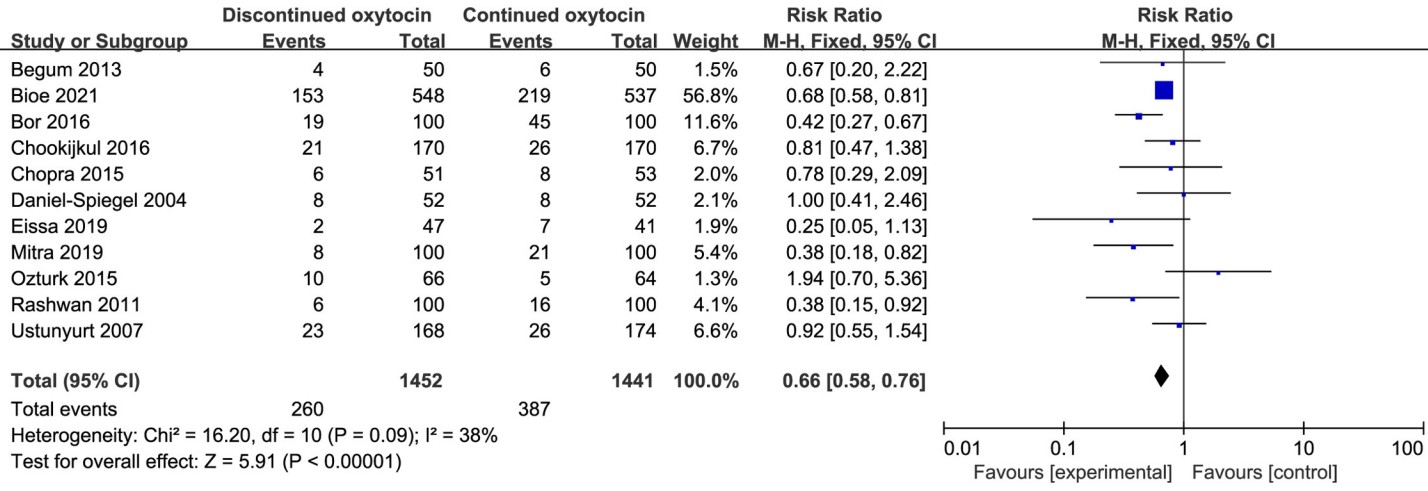

**Fig 10. Pooled results for each outcome.**

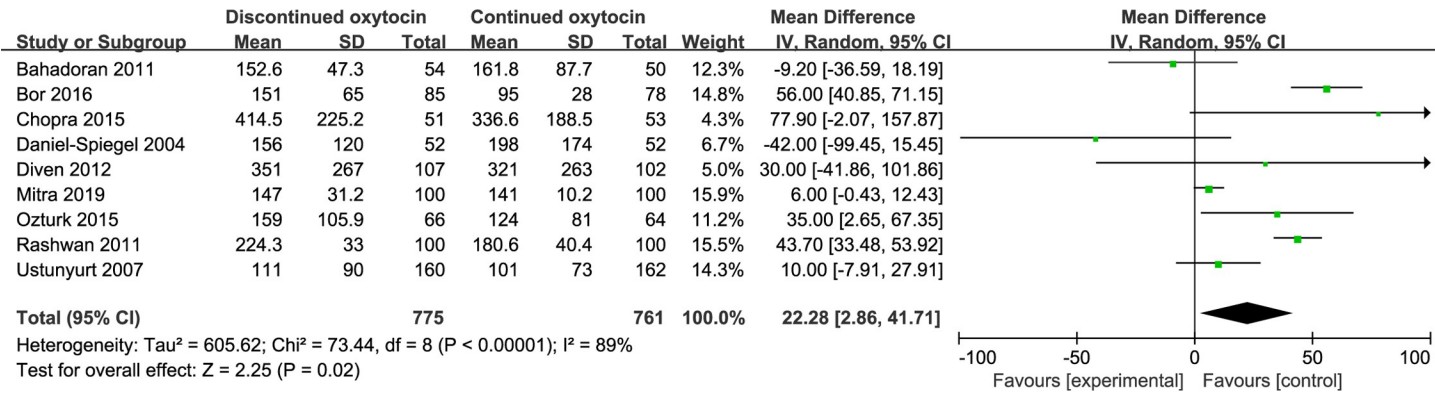

**Fig 11. Pooled results for each outcome.**

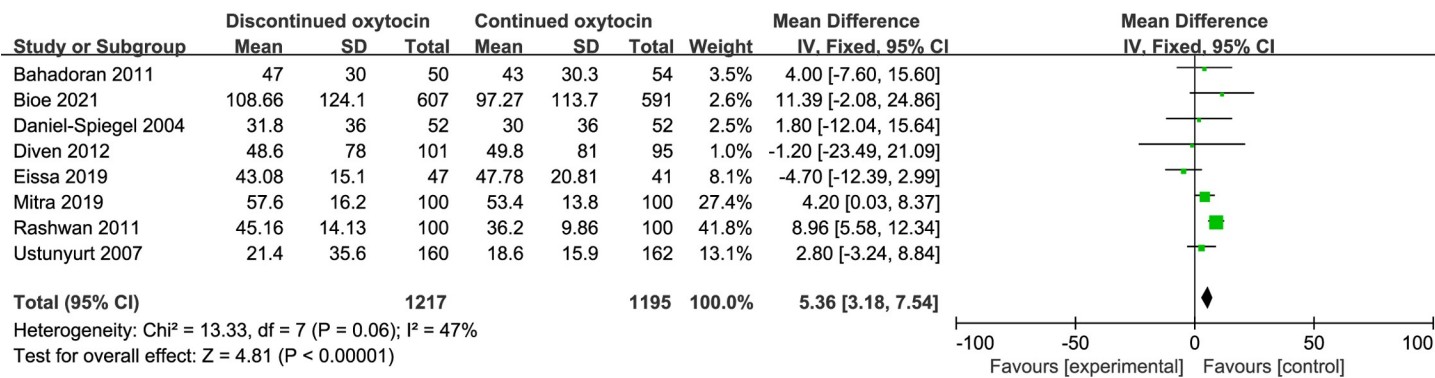

**Fig 12. Pooled results for each outcome.**

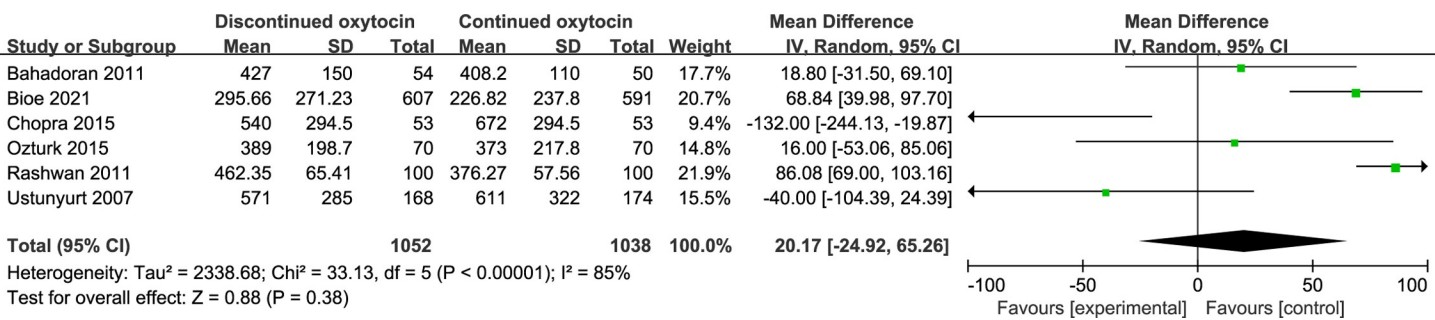

| Study or Subgroup | Discontinued oxytocin | | | Continued oxytocin | | | Weight | Mean Difference IV, Random, 95% CI |
|---|---|---|---|---|---|---|---|---|
| | Mean | SD | Total | Mean | SD | Total | | |
| Bahadoran 2011 | 427 | 150 | 54 | 408.2 | 110 | 50 | 17.7% | 18.80 [-31.50, 69.10] |
| Bioe 2021 | 295.66 | 271.23 | 607 | 226.82 | 237.8 | 591 | 20.7% | 68.84 [39.98, 97.70] |
| Chopra 2015 | 540 | 294.5 | 53 | 672 | 294.5 | 53 | 9.4% | -132.00 [-244.13, -19.87] |
| Ozturk 2015 | 389 | 198.7 | 70 | 373 | 217.8 | 70 | 14.8% | 16.00 [-53.06, 85.06] |
| Rashwan 2011 | 462.35 | 65.41 | 100 | 376.27 | 57.56 | 100 | 21.9% | 86.08 [69.00, 103.16] |
| Ustunyurt 2007 | 571 | 285 | 168 | 611 | 322 | 174 | 15.5% | -40.00 [-104.39, 24.39] |
| | | | | | | | | |
| Total (95% CI) | | | 1052 | | | 1038 | 100.0% | 20.17 [-24.92, 65.26] |

Heterogeneity: Tau² = 2338.68; Chi² = 33.13, df = 5 (P < 0.00001); I² = 85%
Test for overall effect: Z = 0.88 (P = 0.38)

**Fig 13. Pooled results for each outcome.**

contractility [10, 42]. Moreover, the process of labor is demonstrated as self-sustaining after the active labor is established [18]. Accordingly, it seems rational to discontinue oxytocin infusion once establishing the active stage. However, there are not enough proof to know whether the oxytocin should be continued or stopped when the active stage is already established.

The result of our meta-analysis indicates that continued oxytocin leads to a decreased duration of active and second phase of labour, while the total delivery time is not different. However, this effect is accompanied by increased adverse outcomes for both maternal and neonatal, including uterine tachysystole, postpartum hemorrhage, and non-reassuring FHR. It might also be related to an increased risk of chorioamnionitis, yet the conclusion should be interpreted with caution. Taken together, continuing oxytocin infusion when active labour is already established might be inappropriate considering the adverse maternal and fetal outcomes. Future research are needed to confirm our conclusion and clarify the necessity of oxytocin discontinuation after the active labor is already established.

## Conclusion

Once the active phase is established, continuation of oxytocin will not accelerate the total duration of labour and seems to lead to an increased risk of uterine tachysystole, postpartum hemorrhage, and non-reassuring FHR. In addition, a possible raised risk for chorioamnionitis is also observed. Therefore, it is the time for considering discontinuing its infusion after the active phase for avoiding those adverse maternal and neonatal outcomes.

## Supporting information

**S1 Checklist. PRISMA 2009 checklist.**
(DOC)

## Author Contributions

**Data curation:** Danni Jiang, Yang Yang, Xinxin Zhang.

**Formal analysis:** Danni Jiang, Yang Yang, Xinxin Zhang.

**Writing – original draft:** Danni Jiang.

**Writing – review & editing:** Xiaocui Nie.

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
