## [Decision Letter · Decision Letter 0]

28 Dec 2021

PONE-D-21-15843Continued versus discontinued oxytocin after the active phase of labor: an updated systematic review and meta-analysis.PLOS ONE

Dear Dr. Nie,

Thank you for submitting your manuscript to PLOS ONE. After careful consideration, we feel that it has merit but does not fully meet PLOS ONE’s publication criteria as it currently stands. Therefore, we invite you to submit a revised version of the manuscript that addresses the points raised during the review process. Please submit your revised manuscript by Feb 11 2022 11:59PM. If you will need more time than this to complete your revisions, please reply to this message or contact the journal office at plosone@plos.org. Please include the following items when submitting your revised manuscript:A rebuttal letter that responds to each point raised by the academic editor and reviewer(s). You should upload this letter as a separate file labeled 'Response to Reviewers'.A marked-up copy of your manuscript that highlights changes made to the original version. You should upload this as a separate file labeled 'Revised Manuscript with Track Changes'.An unmarked version of your revised paper without tracked changes. You should upload this as a separate file labeled 'Manuscript'.

We look forward to receiving your revised manuscript.

Kind regards,

Giovanni Delli Carpini

Academic Editor

PLOS ONE

Journal Requirements:

Reviewers' comments:

Reviewer's Responses to Questions

**Comments to the Author**

1. Is the manuscript technically sound, and do the data support the conclusions?

Reviewer #1: Partly

Reviewer #2: Yes

2. Has the statistical analysis been performed appropriately and rigorously? 

Reviewer #1: N/A

Reviewer #2: Yes

3. Have the authors made all data underlying the findings in their manuscript fully available?

Reviewer #1: Yes

Reviewer #2: Yes

4. Is the manuscript presented in an intelligible fashion and written in standard English?

Reviewer #1: Yes

Reviewer #2: Yes

5. Review Comments to the Author

Reviewer #1: The regimens of oxytocin during labour vary widely between different guidelines or even individual medical centers, that is why this study about oxytocin administration beyond the active phase is interesting.

Please, provide an explanation for the following comments:

1- If oxytocin discontinuing could decrease the risk of cesarean delivery, how this effect decreased in women who had cesarean operation? Is that affected by other factors like previous cesarean section or other factors that require cesarean intervention?

2- Oxytocin is a drug, regarding dosing protocol: would not different doses result in different responses?

3- The included studies reported different inclusion criteria for the patients, which in turn can affect the analysis?

4- Chorioamnionitis is analyzed, but some patients were enrolled with PROM. How could these affect the result?

5- In table 5, provide a row / column with the total number of events of each arm of the study?

6- Many studies have multiple high risk of bias points, How such studies can be trusted to get an accurate and reliable analysis?

Reviewer #2: Overall, this is an interesting relevant research question and represent a valuable addition to the Labor management approaches. The authors have collected sufficient dataset from the eligible trials and analyzed relevant variables that directly linked to the handled question. The analysis and results interpretation were clear, direct, and illustrated in a well-organized structure. The discussion structure and sequence were enriching and highlighted the important issues related to the study results.

The objective is clearly stated through informative title reflect the true nature of the review; page (1) and the well- structured introduction context; page (2, 3). The methodology was detailed and suitable for the research idea. Performing manual search of references (page (3) method section, Literature search paragraph, last sentence) and Publication bias assessment using the GRADE approach (page (4) method section, Data abstraction, second paragraph, third sentence), are thought to reinforce the design and the quality assessment, respectively. The results were clearly illustrated and neutrally discussed, with useful summary in the pooled outcomes; table (6) page (14) and the analysis was sufficient as well. The authors discussed the oxytocin effect using not just maternal outcomes but also neonatal and delivery interval as well which provide more accuracy to the conclusion; page (16) paragraph 3.3 and 3.4. The thorough discussion was well-structured and handled the relevant literature with results implication along with the existed conflict between using Continued vs discontinued oxytocin based on the available evidence (page 17, paragraph 3). Figures and tables were clearly presented and correctly labelled.

Minor issues:

1) In Method section in Data abstraction part, page (4) first paragraph; the author mentioned an extracted variable (setting, mentioned as a part of General information) that was not obtained in the characteristics table; page (7).

2) In Result section, there are two mistakes in the reported data compared with the labelled analysis figures.

- page (13), Maternal outcomes paragraph, Fifth sentence: ''Moreover, we found a possible increased risk of chorioamnionitis in the OD group without significant heterogeneity [RR (95% CI): 2.77 (1.02–5.08), P=0.05] (Fig 5e)''. RR is supposed to be 2.27 as shown in fig 5e. Also, the number of trials from which these results were analyzed and the number of patients are recommended to mentioned in this sentence.

- Page (16), Induction delivery interval paragraph, second sentence: ''The pooled result indicated that discontinuation may prolong the active stage of labor compared to continued oxytocin [MD (95% CI): 2.28 (2.86–41.71), P=0.02]''. RR is supposed to be 22.28 as shown in fig 5g.

3) In Discussion Section in Comparison and Implication parts (Page 17 and 18, respectively), there was a lack of reported statistical data of the results and conclusions discussed from the literature. The statistical data are important and reinforce the flow of the discussion.

In brief, the paper is generally well thought out and written, accessible data statistically analyzed in sufficient detailed manner, the conclusions well supported by the data presented in the results and figures, and with no conflict of interests. Thus, we recommend its acceptance, with some minor corrections regarding the issues previously mentioned.

6. PLOS authors have the option to publish the peer review history of their article (what does this mean?). If published, this will include your full peer review and any attached files.

Reviewer #1: No

Reviewer #2: **Yes: **Gena M Elassall

---

## [Author Response · Author response to Decision Letter 0]

4 Feb 2022

Response to reviewers

Reviewer #1: The regimens of oxytocin during labour vary widely between different guidelines or even individual medical centers, that is why this study about oxytocin administration beyond the active phase is interesting. Please, provide an explanation for the following comments.

Dear reviewer#1:

 Thank you for your comments concerning our manuscript entitled “Continued versus discontinued oxytocin after the active phase of labor: An updated systematic review and meta-analysis”. Those comments are all valuable and very helpful for revising and improving our paper, as well as the important guiding significance to our researches. We have studied these comments carefully and tried our best to revise and imporve the manuscript. We sincerely hope that it will meet with approval. 

Response to the comments:

1- If oxytocin discontinuing could decrease the risk of cesarean delivery, how this effect decreased in women who had cesarean operation? Is that affected by other factors like previous cesarean section or other factors that require cesarean intervention?

Response: Thanks very much for your valuable comment. Women with a history of prior cesarean delivery were excluded in seven trials (Begum 2013; Chookijkul 2016; Chopra 2015; Diven 2012; Eissa 2019; Mitra 2019; Ustunyurt 2007). For the remnant six studies, they may include patients with previous cesarian delivery, but specific data were limited. So a separate analysis just on the history of cesarian delivery is not possible. We sincerely hope that, with further studies with a larger sample size that stratify by previous cesarean section, can deepen our understanding to this issue. And other factors that require cesarean intervention were already assessed by the authors when performing the study, the exact items are in the exclusion criteria of each study. Since we do not present with the exclusion criteria of 13 studies, we are sorry for confusing you. Thanks again for your careful comment.

2- Oxytocin is a drug, regarding dosing protocol: would not different doses result in different responses?

Response: Thanks for your comment. The drug protocols are somewhat different, but for all trials the procedure corresponded to a low-dose regimen, and this makes the impact to some extent mild across studies (Budden 2014). Though oxytocin has been used for induction of labor for many years, there is no consensus regarding the ideal dosing regimen. Thus, it is impossible to unify the usage dose in all trials around the world. Actually, we totally agree with your comment that different doses may have different responses, and we have discussed this content in 'Strengths and limitations' part. And we do hope, a consensus for oxytocin use may exist as soon as possible with the continuous progress of research in this field.

Reference: Budden A, Chen LJ, Henry A. High-dose versus low-dose oxytocin infusion regimens for induction of labour at term. Cochrane Database Syst Rev. 2014;(10):CD009701. doi:10.1002/14651858.CD009701.pub2

3- The included studies reported different inclusion criteria for the patients, which in turn can affect the analysis?

Response: The inclusion criteria for each trial is somewhat, but they all correspond to the application indication for induction of labor. Since the patients in each hospital are different, it is impossible to include exactly the same patients for every trial. And we do have the same worry as you that different inclusion criteria may affect the analysis, so we discussed this issue in the 'Strengths and limitations' part. In the furture, large multi-center research may help further answer this question.

4- Chorioamnionitis is analyzed, but some patients were enrolled with PROM. How could these affect the result?

Response: Thanks for your comment. There are 2 of the total 13 studies that included patients with PROM （Bioe 2021; Daniel-Spiegel 2004）. Since the primary results were not separately reported according to whether patients had PROM or not, it is not possible for us to do a separate analysis. Your opinion also inspired us a lot. As the application of oxytocin in the induction of labor becomes more and more clear, it may be beneficial to carry out such research in the PROM subgroup in the future for exploring the potential effect of oxytocin in chorioamnionitis. Thanks again for your hard work on our study.

5- In table 5, provide a row / column with the total number of events of each arm of the study?

Response: Thank you for your comment. We totally agree with your opinion and had added a column with the total number of events according to your suggestion, see Table 5. Thanks for your warm work on our article.

6- Many studies have multiple high risk of bias points, How such studies can be trusted to get an accurate and reliable analysis?

Response: Thanks for the comment. We examined the quality of the overall body of the evidence for the outcomes using GRADE. As you said, our GRADE assessments ranged from very low certainty to moderate certainty. Overall, some of the included trials had some design limitations. Luckily, we found high consistency for our primary outcome, cesarean delivery rate. And this, indicates a highly reliable conclusion. However, the effects on other outcomes should be interpreted with caution. And in the future, further high-quality studies are needed to confirm our conclusion. Appreciate your warm work on this article earnestly.

 Again, we appreciate your warm work earnestly and thank you very much for your comments and suggestions to our paper. All the authors have studied your comments carefully, we think it is important guidance to our paper. Exploring the optimal scheme for the management of induced labor is of vital importance for improving health outcomes for both women and infants. Though extensively used, its ideal regimen regarding initial dose, increments, and maximal dose of oxytocin is still a lack of consensus. In recent years, discontinuing oxytocin infusion when the active phase of labor is reached gains much attention. However, there is not enough proof to know whether the oxytocin should be continued or stopped when the active stage is already established. As you said, there are some design limitations of studies on this topic, and we do hope to poll the related data and decrease this bias by expanding the sample size through the meta-analysis. In the future, large multi-center research may help further answer this question. Once again, thank you for your hard work during this special time of the COVID-19 pandemic, hope all is well with you and your family.

Thank you and best regards.

Sincerely yours,

Nie Xiaocui

Reviewer #2: Overall, this is an interesting relevant research question and represent a valuable addition to the Labor management approaches. The authors have collected sufficient dataset from the eligible trials and analyzed relevant variables that directly linked to the handled question. The analysis and results interpretation were clear, direct, and illustrated in a well-organized structure. The discussion structure and sequence were enriching and highlighted the important issues related to the study results.

The objective is clearly stated through informative title reflect the true nature of the review; page (1) and the well- structured introduction context; page (2, 3). The methodology was detailed and suitable for the research idea. Performing manual search of references (page (3) method section, Literature search paragraph, last sentence) and Publication bias assessment using the GRADE approach (page (4) method section, Data abstraction, second paragraph, third sentence), are thought to reinforce the design and the quality assessment, respectively. The results were clearly illustrated and neutrally discussed, with useful summary in the pooled outcomes; table (6) page (14) and the analysis was sufficient as well. The authors discussed the oxytocin effect using not just maternal outcomes but also neonatal and delivery interval as well which provide more accuracy to the conclusion; page (16) paragraph 3.3 and 3.4. The thorough discussion was well-structured and handled the relevant literature with results implication along with the existed conflict between using Continued vs discontinued oxytocin based on the available evidence (page 17, paragraph 3). Figures and tables were clearly presented and correctly labelled.

Dear reviewer#2:

 Thank you very much for your comments concerning our manuscript entitled “Continued versus discontinued oxytocin after the active phase of labor: An updated systematic review and meta-analysis”. Those comments are all valuable and very helpful for revising and improving our paper. We have studied these comments carefully and tried our best to revise and imporve the manuscript. We sincerely hope that it will meet with approval. Again, we appreciate your positive evaluation of our work. Thank you for your hard work in this difficult time of the COVID-19 pandemic, hope all is well with you and your family.

Response to the comments:

1) In Method section in Data abstraction part, page (4) first paragraph; the author mentioned an extracted variable (setting, mentioned as a part of General information) that was not obtained in the characteristics table; page (7).

Response: Thanks for your comment. The ‘setting’ we mentioned means the background of the study, it includes the author, publication year, and country of the study. We are so sorry that our inaccurate word confused you and we have deleted this word from the ‘Data abstraction part’. Thanks again.

2) In Result section, there are two mistakes in the reported data compared with the labelled analysis figures.

- page (13), Maternal outcomes paragraph, Fifth sentence: ''Moreover, we found a possible increased risk of chorioamnionitis in the OD group without significant heterogeneity [RR (95% CI): 2.77 (1.02–5.08), P=0.05] (Fig 5e)''. RR is supposed to be 2.27 as shown in fig 5e. Also, the number of trials from which these results were analyzed and the number of patients are recommended to mentioned in this sentence.

- Page (16), Induction delivery interval paragraph, second sentence: ''The pooled result indicated that discontinuation may prolong the active stage of labor compared to continued oxytocin [MD (95% CI): 2.28 (2.86–41.71), P=0.02]''. RR is supposed to be 22.28 as shown in fig 5g.

Response: Thank you very much for your careful comment. We are so sorry for the mistake and made corrections in accordance with your comment. Also, we added the number of trials from which these results were analyzed in the sentence. Thanks.

3) In Discussion Section in Comparison and Implication parts (Page 17 and 18, respectively), there was a lack of reported statistical data of the results and conclusions discussed from the literature. The statistical data are important and reinforce the flow of the discussion.

Response: Thanks for your comment. We totally agree with your comment that statistical data are very important and we had put them in the initial version of our manuscript. However, when we submitted the manuscript to the journal, the 'Instruction for authors' part suggests we delete the repeat comment to make the article concise and readable. So deleted this content when we submitted the article since all the statistical data had been presented in Table 6.

 We tried our best to improve the manuscript and made some changes in the manuscript according to your comments. We hope the corrections will meet with approval. Again, we appreciate your warm work earnestly during this hard time of COVID-19, and we hope all is good for you and your family. Looking forward to hearing from you.

Thank you and best regards.

Sincerely yours,

Nie Xiaocui

---

## [Decision Letter · Decision Letter 1]

11 Apr 2022

Continued versus discontinued oxytocin after the active phase of labor: an updated systematic review and meta-analysis.

PONE-D-21-15843R1

Dear Dr. Nie,

We’re pleased to inform you that your manuscript has been judged scientifically suitable for publication and will be formally accepted for publication once it meets all outstanding technical requirements.

Kind regards,

Giovanni Delli Carpini

Academic Editor

PLOS ONE

Reviewers' comments:

Reviewer's Responses to Questions

**Comments to the Author**

1. If the authors have adequately addressed your comments raised in a previous round of review and you feel that this manuscript is now acceptable for publication, you may indicate that here to bypass the “Comments to the Author” section, enter your conflict of interest statement in the “Confidential to Editor” section, and submit your "Accept" recommendation.

Reviewer #1: All comments have been addressed

2. Is the manuscript technically sound, and do the data support the conclusions?

Reviewer #1: Yes

3. Has the statistical analysis been performed appropriately and rigorously? 

Reviewer #1: Yes

4. Have the authors made all data underlying the findings in their manuscript fully available?

Reviewer #1: Yes

5. Is the manuscript presented in an intelligible fashion and written in standard English?

Reviewer #1: Yes

6. Review Comments to the Author

Reviewer #1: (No Response)

7. PLOS authors have the option to publish the peer review history of their article (what does this mean?). If published, this will include your full peer review and any attached files.

Reviewer #1: No

---

## [Editor Report · Acceptance letter]

22 Apr 2022

PONE-D-21-15843R1 

Continued versus discontinued oxytocin after the active phase of labor: An updated systematic review and meta-analysis. 

Dear Dr. Nie:

I'm pleased to inform you that your manuscript has been deemed suitable for publication in PLOS ONE. Congratulations! Your manuscript is now with our production department. 

Kind regards, 

on behalf of

Dr. Giovanni Delli Carpini 

Academic Editor

PLOS ONE